# *Streptococcus suis* Isolates—Serotypes and Susceptibility to Antimicrobials in Terms of Their Use on Selected Repopulated Czech Pig Farms

**DOI:** 10.3390/pathogens10101314

**Published:** 2021-10-13

**Authors:** Ján Matiašovic, Kateřina Nedbalcová, Marek Žižlavský, Petr Fleischer, Lucie Pokludová, Dita Kellnerová, Kateřina Nechvátalová, Bronislav Šimek, Linda Czanderlová, Monika Zouharová, Jan Bernardy, Natálie Králová, Soňa Šlosárková

**Affiliations:** 1Department of Infectious Diseases and Preventive Medicine, Veterinary Research Institute, 62100 Brno, Czech Republic; nedbalcova@vri.cz (K.N.); fleischer@vri.cz (P.F.); nechvatalova@vri.cz (K.N.); zouharova.m@vri.cz (M.Z.); bernardy@vri.cz (J.B.); kralova.n@vri.cz (N.K.); slosarkova@vri.cz (S.Š.); 2Sevaron s.r.o., 61200 Brno, Czech Republic; marek@sevaron.cz (M.Ž.); laborator@sevaron.cz (D.K.); linda@sevaron.cz (L.C.); 3Department of Marketing Authorisation, Institute for State Control of Veterinary Biologicals and Medicines, 62100 Brno, Czech Republic; pokludova@uskvbl.cz; 4Department of Molecular Biology, State Veterinary Institute Jihlava, 58601 Jihlava, Czech Republic; simek@svujihlava.cz; 5Department of Experimental Biology, Faculty of Science, Masaryk University, 62500 Brno, Czech Republic

**Keywords:** *Streptococcus suis*, SPF farm, serotype, MLST, antimicrobial susceptibility, antimicrobial treatment

## Abstract

*Streptococcus suis* represents a primary health problem (such as meningitis, septicemia and arthritis in piglets and fatteners) in the swine industry worldwide and also an emerging zoonotic pathogen. In the Czech Republic, many pig farms repopulated their herds over the past decades to reduce morbidity and minimize treatment. The study analysed serotypes, sequence types and antimicrobial susceptibility in 39 *S. suis* isolates obtained from organs of diseased pigs from selected 16 repopulated farms with a history of *S. suis*-associated diseases and routine antimicrobial treatment with tulathromycin and/or amoxicillin. The analysis revealed diversity of collected isolates with regular occurrence of more than three serotypes per farm. The serotypes identified were 1/2 and 7, each in six isolates, followed by serotype 2 and 3 found in five isolates each, other serotypes were less frequent. Seven isolates were not typable by multiplex PCR and we also found sequence type of unknown type in thirteen isolates. The majority of *S. suis* isolates were resistant to clindamycin (n = 31), tetracycline (n = 29) and tilmicosin and tulathromycin (n = 28). On the other hand, with the exception of two isolates that were intermediately susceptible to penicillin and ampicillin, all isolates were susceptible to all three tested subgroups of beta-lactam antibiotics.

## 1. Introduction

*Streptococcus suis* (*S. suis*) is a causative agent of important diseases, particularly in pigs [1,2], but it can also cause infections in humans and other animal species. *S. suis* infections currently occur in pig farms worldwide and are responsible for diseases that often result in the death of affected animals, especially weaned piglets, with a negative impact on the economic situation of the farm [3]. According to the survey performed by the Federation of Veterinarians of Europe (FVE), *S. suis* belongs among five pathogens of weaners, where the largest amounts of antimicrobials are used in pigs [4].

Based on the differences in capsular polysaccharides (CPS), 35 different *S. suis* serotypes (1–34 and 1/2) were successively described [5,6,7,8]. Later, some serotypes were classified as other bacterial species based on their genomic analysis [9,10]. Therefore, we currently determine only 29 serotypes of *S. suis* [11]. Serotype information is very important from an epidemiological point of view, as some serotypes have the ability to cause more serious disease than other serotypes. Different serotypes occur in different parts of the world, which plays a role in assessing the effectiveness of the use of vaccines that contain inactivated strains of certain *S. suis* serotypes [3]. The published studies indicated that, in general, serotypes 1/2, 2, 3, 7 and 9 are most often isolated from diseased pigs, but a relatively high percentage of *S. suis* isolates remains serologically nontypable [3,12].

However, by standard serological testing or by later developed specific PCRs to detect genes encoding the CPS production of individual serotypes [11,13], serotype 2 cannot be clearly distinguished from serotype 1/2 and also serotype 1 from serotype 14 because they cross-react with each other [11,13,14]. It is very important because serotypes 2 and 14 are associated with human diseases [15]. Whole-genome sequencing has shown that the gene content at the CPS locus for serotype 1 and for serotype 14 is the same and similarly CPS locus for serotype 2 is the same as CPS locus for serotypes 2 and 1/2, but a single nucleotide polymorphism in the *cpsK* gene (G for T or C) was revealed [16]. All serotype 2 and 14 isolates in that study had nucleotide G at position 483 of the *cpsK* gene, while all isolates of serotypes 1 and 1/2 had nucleotide C or T at this position. Therefore, detection of this nucleotide in the *cpsK* gene makes it possible to distinguish serotypes 2 from 1/2 or 1 from 14. However, a single-nucleotide G→C/T substitution at nucleotide position 483 in the *cpsK* gene causes preferential addition of galactose (in the case of serotypes 2 and 14) or N–acetylgalactosamine residues (in the case of serotypes 1/2 and 1) to the capsular polysaccharide repeating units [17]. Identification of this mutation by reading the *cpsK* sequence or by a recently published method based on high-resolution melting analysis [18] enables specific identification of these serotypes. Recently, a simple and low-cost method of detecting *cpsK* gene polymorphism based on PCR-restriction fragment length polymorphism (PCR-RFLP) was presented [19].

*S. suis* gains entrance to the susceptible piglets after birth and young pigs through the oropharyngeal mucosa and is carried in the tonsils, nasal mucosa, and mandibular lymph nodes of healthy animals, particularly in survivors of an outbreak [20]. Due to the young age of the piglets affected and the lack of an effective vaccine, many farms use metaphylactic perinatal antimicrobials to control *S. suis* disease [21]. A similar situation is with the diseases produced by other early colonisers of piglets, such as *Haemophilus parasuis* and *Mycoplasma hyorhinis* [21].

If preventive and control measures are not sufficient and *S. suis* infection breaks out, early initiation of parenteral treatment of sick pigs with appropriate antimicrobial agents is most important for the success of therapy. The drugs of choice are the penicillin group of antimicrobials, especially broad-spectrum beta-lactams such as ampicillin or amoxicillin. Although penicillins are generally effective against *S. suis* infections, the detection of strains resistant to these antimicrobials has been published. Resistances to these antimicrobials were found to varying degrees in published studies and national monitoring programmes, mostly from 0 to 25%. Therefore, the effective antimicrobial treatment should be initiated only after the susceptibility or resistance of the current isolate to antimicrobials has been determined. The use of a combination of trimethoprim and sulfamethoxazole is further contemplated for the treatment of *S. suis* infections. Relatively high levels of resistance of *S. suis* isolates to macrolides, lincosamides, tetracyclines or sulfonamides were reported from many countries [22]. The basic source of information for the considered administration of antimicrobials is regular antimicrobial susceptibility testing (AST) of bacterial pathogens [2,12,22]. In the Czech Republic, national antimicrobial resistance monitoring [23,24,25,26] of target pathogens, including *S. suis*, has been taking place since 2016. It includes isolates obtained at the State Veterinary Institutes (SVU) by testing samples from diseased pigs from any farm in the Czech Republic, which meet the specified criteria [23,24,25,26].

Management techniques that integrate technology into disease control, elimination and eradication have been further levelled up by repopulating the entire herd by eliminating serious infections caused by *A. pleuropneumoniae*, etc., to avoid further medication and cost increases [27]. A more detailed analysis of *S. suis* characteristics in repopulated specific pathogen free (SPF) herds in Central Europe has not yet been carried out.

The aim of our study was to analyze the serotypes, including multilocus sequence typing (MLST) and antimicrobial susceptibility, in *S. suis* isolates obtained from diseased pigs from repopulated SPF farms selected on the basis of *S. suis* occurrence taking into account the routine antimicrobial control treatments of bacterial infections used on them.

## 2. Results

### 2.1. Serotyping and Multilocus Sequence Typing (MLST)

Out of 39 isolates, serotype was not determined in seven cases (Figure 1). Of these seven isolates, each isolate originated from a different farm; six of them were isolated from the lungs, one from the brain (Table 1). The serotypes identified were 1/2 and 7, in six isolates each, followed by serotype 2 and 3 found in five isolates each. Other four serotypes were less frequent (Figure 1).

MLST revealed relatedness of isolates of the same serotype or clonal complex of the same ST but different serotype (Figure 2).

Although serotype 1/2 was found in three different and distant farms (A, I, P), all six isolates are of the same ST28 (Table 1). It suggests they are closely related. Similarly, all six S7 (serotype 7) isolates originating from three farms (A, B, J) were of ST29, indicating a close relationship of these isolates.

On the other hand, five isolates of serotype 2 originating from four farms (A, E, H, M) were of three different STs. S2 ST1, a well-known highly virulent type with zoonotic potential, was found in two different farms (E, M). Two other S2 isolates were of ST54-like, the last one was of ST28, indicating genetic diversity of S2 isolates collected.

Five serotype 3 isolates originated from four farms (B, D, G, K) and were of ST54 except one isolate similar to ST1521 or 1546, but having aroA 8 allele. Similarly, three out of four S8 isolates were of ST87, but one was ST1546. Interestingly, none of the three S16s was of known ST. Each out of two S23 isolates was from a different farm (B, D) and STs of both have not yet been described.

The only S31 isolate was of unknown ST. Other isolates were not typable by multiplex PCR method. Five were of known STs but the other two were of not yet described STs.

### 2.2. Antimicrobial Susceptibility Testing

Using a microdilution method, MICs of the tested antimicrobials were determined (Table 1) for all 39 *S. suis* isolates. The numbers of susceptible, intermediate and resistant isolates to individual antimicrobials and to combinations of antimicrobials are shown in Figure 3. All isolates were susceptible to ceftiofur, enrofloxacin, florfenicol and to both tested combinations of antimicrobials (amoxicillin/clavulanic acid and trimethoprim/sulfamethoxazole). Only two isolates (farm O) were intermediate to ampicillin and penicillin (remaining isolates were susceptible). On the other hand, most of the tested isolates were resistant to clindamycin (31 resistant isolates, 8 susceptible isolates), to tetracycline (29 resistant isolates, 10 susceptible isolates) and 28 isolates were resistant and 11 isolates were susceptible to tilmicosin and tulathromycin. To tiamulin, 21 isolates were susceptible, 6 isolates were intermediate and 12 isolates were resistant. Distribution of MICs of isolates in the tested antimicrobials is shown in Table 1.

Only two isolates (G2 and G5) were susceptible to all tested antimicrobials. According to the profiles of phenotypic resistance, 27 isolates were considered as multidrug resistant (isolates resistant to three or more groups of antimicrobials). The profiles of phenotypic resistance are listed in Table 2.

The profiles of phenotypic resistance (five antimicrobials with resistance) quite widely varied on four farms (A, B, D, G) with more than two isolates (i.e., with four or more isolates). On these farms, the resistance to clindamycin varied from two out of four (2/4) isolates (farm B) to all isolates (farms A, D); to tetracycline from 2/5 (G) to 8/9 (A); to both macrolides from 1/5 (G), to all isolates (farms A, D); and to tiamulin from 1/5 (G) to 3/9 (A), i.e., farm G had minimum resistance to four of these five antimicrobials (Table 1). Multidrug resistance was identified on these four farms in 4/4 (D), 8/9 (A), 2/4 (B) and 1/5 (G) *S. suis* isolates (Table 3).

### 2.3. Routine Antimicrobial Control Treatments and Resistance on Farms

On farms, which used tulathromycin for routine perinatal treatment (Table 3), 23/30 (76.7%) of *S. suis* isolates were resistant to it, while on farms without such a routine it was numerically less: 5/9 (55.6%). Resistance to tulathromycin (and always to tilmicosin, i.e., macrolides) closely coincided with multidrug resistance (agreement in 36/39, i.e., 92.3% of isolates); only three isolates did not match in this respect (Table 2): two were resistant to tulathromycin but were not multidrug resistant and, in contrast, only one was multidrug-resistant, although not resistant to tulathromycin. No *S. suis* isolate was resistant to any of the four beta-lactam antibiotics tested.

## 3. Discussion

Among thirty-nine isolates of *S. suis* obtained from clinical samples of diseased pigs from fifteen pig farms located in the Czech Republic (CZ) and one farm in Slovakia, a number of different serotypes and STs were found. Seven isolates were not typable by the multiplex PCR used [27] and also STs of unknown type was found in thirteen isolates. Curiously, none of the three isolates of serotype 16 had known ST. However, this wide genetic variability is not surprising, because *S. suis* is a genetically highly divergent bacterium with currently recognized 29 serotypes [28] and 1620 ST described up to the end of June 2021 (pubmlst.org *S. suis* profile scheme). Serotype 2 isolates collected in our study were not ST uniform but among five S2 isolates, three different STs were found. Besides the well-known S2 ST1 [29,30], also one S2 ST28 isolate belonging to clonal complex CC28 [31] was found. Interestingly, on one farm, two S2 isolates with yet unknown ST similar to ST54 were identified. This new ST differs from ST54 by the presence of the aroA1 allele. Both isolates originated from the brain of diseased pigs, suggesting they could be clinically relevant.

The presence of three or more different *S. suis* serotypes per farm was detected in all four farms, from which four or more *S. suis* isolates were available. It well illustrates ecology of this bacterium exploiting different niches [32] and it is an opportunity for the development of genetic plasticity, such as serotype switching [33].

On the other hand, most prevalent serotypes found in our study, i.e., serotype 1/2 and 7, were each of the same ST. Although the isolates of both serotypes were found on at least three farms, it suggests a possible common source of these serotypes.

The last publicly available data on the average total consumption of antimicrobials on Czech pig farms showed 122.7 mg per population correction unit [34] (the data on sales and population of pigs in 2016). From 2016 to 2020 (CZ), the total consumption of antimicrobials dropped significantly. The total consumption of antimicrobials (AM) on the farms studied seems lower than the state average for the CZ, as they routinely used antimicrobials perinataly to control early colonizers of piglets (newborn piglets; injectable, i.e., a small volume of tulathromycin) and at the weaning period (a small total volume of amoxicillin used in piglets compared to older animals). The presumption that antimicrobial consumption on the farms under study was lower than the average in the Czech Republic is based on the fact that oral group medication in older animals in later pre-fattening or later fattening was rarely used there, because these are repopulated farms with elimination of serious infections caused by *A. pleuropneumonie* etc., i.e., infections associated with frequent antimicrobial treatment [4]. Moreover, it can be reasonably assumed that in older pigs, antimicrobials are most commonly administered orally via medicated feed or water, which leads to significantly higher volumes of antimicrobials consumed.

Routine one-shot injectable tulathromycin and/or short-term aminopenicillin administrations are believed/experienced to help to reduce the incidence of the infections of both respiratory and gastrointestinal tract as well as *S. suis* infections. Such approach is considered (especially by farmers) as beneficial due to decreasing morbidity and mortality, with final economic profit and minimizing the costs due to necessity of the treatment of pigs of higher weight categories in the later stage of fattening. Hovewer this approach is controversial from the current perspective of responsible use of antimicrobials. It is associated with the issue of labour costs and human resources, economic demands as well as the difficulties with change of the behavioural frameworks.

Monitoring of *S. suis* antimicrobial resistance is carried out worldwide, especially in countries with intensive pig production, including CZ. However, although this pathogen causes serious illness in pigs (as well as in humans) worldwide, there is no internationally recognized summary of clinical breakpoints of antimicrobial resistance for all antimicrobials with clinical importance for this microorganism. Therefore, some breakpoints had to be taken or derived from clinical breakpoints established for other animals, humans or other (similar) pathogens. It also complicates the comparison of results from different studies, as different reports of antimicrobial resistance use different clinical breakpoints [35], especially when there are not available MICs, but just data on prevalence of susceptibility/resistance. Our study does not aim to present an overview of *S. suis* resistances in the CZ. Thirty-nine selected *S. suis* isolates from the repopulated pig farms were typed and characterized by various methods. The results of antimicrobial susceptibility testing of individual isolates are one of the selected indicators in the description of the tested isolates.

Antimicrobials that are commonly used to treat *S. suis* infections (penicillin, amoxicillin) were effective in most cases in our study. Surprisingly, despite the worldwide use of beta-lactams in pigs for over 50 years, the majority of clinical *S. suis* remains sensitive to these antibiotics [21]. Two of our isolates were intermediate to penicillin and ampicillin. This study showed, as for resistance to penicillins, more favourable results than the Czech National Antibiotic Programme (NAP) monitoring, even though the same diagnostic kits manufactured by the co-authors at the Veterinary Research Institute and the same clinical breakpoints were used in both studies. Within the NAP, AST was performed on other 56 isolates in 2020 and PNC resistance was observed in 8.9%, but no intermediate susceptibility was detected [26]. According to previously published data from Spain and Brazil, only low percentages of the tested *S. suis* isolates were intermediate or resistant to penicillin antimicrobials [12,36,37,38] and are primarily found in commensal sites [21]. Beta-lactam resistant strains could spread if the susceptible (clinical) strains were wiped out [21]. The isolates resistant to penicillin antimicrobials were also resistant to other types of antimicrobials in many cases [38]. The choice of antimicrobials to treat such an infection can then be problematic.

The third generation of cephalosporins (ceftiofur) and fluoroquinolones (enrofloxacin), which are classified according to the AMEG recommendation as Category B—“Restrict”, has been the most effective substance for *S. suis* infections so far [39]. Unfortunately, resistances to this antimicrobial have already been recorded [12,35,37] and rarely occur also in the CZ [23,24,25,26]. However, all isolates were susceptible to these antimicrobials in our study.

The combined antimicrobials (trimethoprim with sulfamethoxazole and amoxicillin with clavulanic acid) showed good susceptibility. We found no resistant or intermediate isolate to these fixed combinations. It is very interesting to compare the effectiveness of this antimicrobial in different parts of the world. While in Europe the incidence of resistant strains to trimethoprim/sulfamethoxazole is relatively low (Spain 0%, The Netherlands 3%, England 12%) [35,36,39], in South America and Asia, very high incidence of resistant strains was found (Brazil 97.7%, Thailand 60%) [37,38].

The isolates in our study showed the greatest resistance to lincosamides (clindamycin), tetracyclines and macrolides (tilmicosin, tulathromycin). High resistance to these antimicrobials has also been reported in other countries, not only in pigs but also in humans [12,35,38]. The high level of resistance to tetracyclines and macrolides could be related to the wide use of these antimicrobials in therapy in veterinary medicine [37] and to all kinds of use, including growth promotion in different parts of the world so far, and treatments/prophylaxis/metaphylaxis of other infectious diseases.

Therefore, these three groups of antimicrobials were completely unsuitable for the therapeutic or control treatments of *S. suis* infections with the exception of both macrolides on farm G (one of the four farms with four or more *S. suis* isolates per farm), where four of five isolates were susceptible to them. Regarding the high diversity of isolates on individual farms, we cannot consider these AMs and tiamulin as suitable for managing *S. suis* infections on farms represented by 1–2 isolates, even if the isolates are susceptible to them.

None of the *S. suis* isolates was resistant to any of the four penicillin antibiotics tested, which indicated little or no association between the short-term administration of amoxicillin to all weaned piglets and the detection of multidrug-resistant isolates. The higher proportion of isolates resistant to tulathromycin (or to both macrolides) in the group of farms routinely using it in all piglets in the first week of life compared to farms without such treatment cannot be directly linked to this administration, especially in view of the fact that the number of isolates included was limited. The proportion of tulathromycin-resistant isolates in our study was significantly higher than that found by the NAP, which recorded 46.4% of tulathromycin-resistant and 55.4% tilmicosin-resistant *S. suis* isolates in the year 2020 [26] and 53.3% of tulathromycin-resistant *S. suis* isolates when compared the years 2017–2020 [23,24,25,26]. This can probably be attributed to the increased use of tulathromycin, either in the monitored farms or in the farms from which breeding gilts were purchased. The striking differences in the proportion of multidrug-resistant isolates between some selected farms using tulathromycin (and amoxicillin) were very biased by low numbers: farm D 4/4 vs. farm G 1/5.

Veterinary medicine should share the public health concerns related to antimicrobial resistance and should take its part of responsibility for prudent use of antimicrobials, especially in food-producing animals, keeping in mind the risks associated with the increase of antimicrobial resistance. An additional problem of antimicrobial usage is that perinatal antimicrobial treatments can also affect the beneficial bacteria of the microbiota [21].

The use of correctly selected antimicrobials is crucial for the determination of procedures for the treatment of infections, including *S. suis* infections [22]. This is possible on the basis of accurate diagnosis of antimicrobial resistances in pathogens. The results allowed targeted therapy by selecting appropriate drugs, and the results of quantitative methods, such as determining minimum inhibitory concentrations of antimicrobials for individual isolates, allow better adjustment of individual drug doses, route of administration and dosing schedules depending on treatment duration.

Regarding *S. suis* infections, it can also be inferred from our study that it is necessary to monitor susceptibility and trends in susceptibility and resistance mainly to the time-proven simpler beta-lactams (PNC and AMO) and, if the health situation depending on the occurrence of other pathogens allows, not to use other AMs very often for a number of reasons, which are briefly discussed above.

## 4. Materials and Methods

A total of 39 *S. suis* isolates coming from 16 repopulated pig farms with known occurrence of this pathogen and collected between January 2020 and February 2021 were tested.

### 4.1. Farms

The samples were collected from 16 farms (designated A–P). In all cases, they were progressive conventional farms, repopulated with SPF herds, which are declared as free from a number of infections and diseases, including *Betaarterivirus suid 1*, formerly porcine reproductive and respiratory syndrome virus (PRRSV), mycoplasma, pleuropneumonia, swine dysentery, atrophic rhinitis, mange and lice. Some of them were SPF +, i.e., with the occurrence of *Mycoplasma hyopneumoniae*. With the exception of one farm (G), they all used Danish genetics Danbred and their principles of farm management. Farm G used genetics (and management) from PIC. Danbred farms/companies B, C and J were closed turnover farms with their own production of breeding material. Farms D and E imported only purebred sows from Denmark. The other farms purchased commercial F1 gilts from Denmark, and only farms F and I purchased F1 gilts from the Czech Republic (F from E). With the exception of one farm (P), which was in the Slovak Republic, all others were located in different parts of the Czech Republic (CZ). Mostly farrow to finish farms were included (14 farms), one farm (E) was specialized only in piglet production (including gilts for breeding purposes for farm F) and one farm (N) represented reared fatteners (pigs from farm E after weaning to finishing period). The indicative numbers of animals of the farm/herd (Table 3) were two up to 500 sows, four up to 1000 sows, three up to 1500 sows and six up to 2000 sows. After exclusion of one farm in Slovakia, the total number of sows kept on these farms was approximately 17,700, which represented 19.8% of the number of sows reared in the Czech Republic in the respective period (https://www.czso.cz/documents/10180/122621601/27013620p103.pdf/20c350f3-cc8b-4f45-9a1f-ff6253331c2f?version=1.1 (accessed on 1 October 2021)). These were 16 farms that actively sought to address the health situation of the herd and the production of an autogenous vaccine against *S. suis*.

### 4.2. Routine Antimicrobial Control Treatments

On the study farms, routine antibiotic control treatments (metaphylaxis) of various bacterial infections have been performed for years (Table 3). Tulathromycin was routinely administered parenterally (one shot) to all piglets in the first week of life on 10 of the 16 study farms. Medication with amoxicillin through drinking water was used in all weaned piglets (25–28 days of life) on 14 farms. Both control treatments were performed on eight farms.

### 4.3. Bacterial Sampling

At each farm, samples were taken from animals with clinical signs of *S. suis* infection, with a history of only the routine antimicrobial control treatments, mostly from weaners (n = 28) and, to a lesser extent, from newborn piglets (n = 7) or fattening pigs (n = 4). The samples were taken by employees of a private company providing veterinary services and laboratory diagnostics to these farms.

The samples (swabs) were obtained from animals euthanized by T61 (Intervet International B.V., Boxmeer, The Netherland) or from animals which had just died (within 6 h after death) as part of a diagnostic autopsy performed directly on the farm from tissues and organs with apparent pathologies.

The swabs were immediately placed in tubes containing liquid Amies transport medium (Merck, Darmstadt, Germany), transported to the laboratory (Sevaron s.r.o., Brno, Czech Republic) at a temperature of 4 °C, and processed for bacteriological analysis within 18 h of sampling.

### 4.4. Bacterial Isolation and Identification

The samples were subjected to conventional bacteriology (cultivation, isolation and identification of the agents) in an accredited diagnostic laboratory according to standard operating procedures. All samples were inoculated onto Blood Agar (Trios Ltd., Prague, Czech Republic) and Staph / Strep selective medium (Columbia CNA agar, Oxoid, Thermo Fisher Scientific, Waltham, MA, USA) and incubated aerobically at 37 ± 1 °C for 18–24 h. Typical colonies were subcultured on blood agar. The isolated pure cultures of *S. suis* were identified by biochemistry and then by mass spectrometry (Bruker Microflex and software Maldi Biotyper 3.0, Database CD BTYP3.0 –Library, Germany; Bruker, Billerica, MA, USA) using MALDI TOF method (Matrix Assisted Laser Desorption / Ionization—Time of Flight). Thirty-nine isolates of *S. suis* (from 39 animals) were obtained predominantly from lungs (n = 20) and brain (n = 10), as well as from heart (n = 3), joints (n = 3), pericardium, eyes, and pleural swabs (Table 1).

### 4.5. PCR Serotyping, Species Verification and Multi Locus Sequence Typing

For serotype determination, multiplex PCR in four separate PCR reactions was carried out according to [40] with some modifications. Each of the 39 isolates tested was derived from a different animal.

The primers targeted the following genes: (i) glycosyltransferase genes *cps1J*, *cps14J*, *cps1/2J*, *cps2J*, *cps3J*, *cps7H*, *cps9H*, *cps16K*, *cps21N*, *cps23I* and *cps24L*; (ii) capsular polysaccharide repeat unit transporter genes *cps3K*, *cps4M* and *cps5N*; (iii) UDP-glucose dehydrogenase gene *cps4N*; (iv) oligosaccharide repeat unit polymerase genes *cps6I*, *cps10M*, *cps11N*, *cps12J*, *cps13L*, *cps15K*, *cps17O*, *cps18N*, *cps19L*, *cps25M*, *cps27K*, *cps28L*, *cps29L*, *cps30I* and *cps31L*; (v) N-acetylmannosaminyltransferase gene *cps8H*; and (vi) glycerophosphotransferase gene *cps25N*. The primers used in this study for serotyping were according to a previous study [41].

The first PCR reaction included the primers for serotypes 1 + 14, 2 + 1/2, 3, 7, 9, 11, 14, 16 and species-specific gene *recN*. This *recN* gene was added for *S. suis* verification. It is specific for *S. suis* and it is not amplified in serotypes (20, 22, 26, 32, 33 and 34), which have recently been excluded from the species *S. suis* after reclassification [42]. The second reaction included the primers for serotypes 4, 5, 8, 12, 18, 19, 24 and 25, the third included the primers for serotypes 6, 10, 13, 15, 17, 23 and 31, and the fourth reaction included the primers for serotypes 21, 27, 28, 29 and 30.

A few colonies of pure bacterial culture were resuspended in 50 µL of sterile distilled water. The suspension was incubated for 10 min at 100 °C and centrifuged for 10 min at 10,000× *g*. The supernatant was used in the PCR reaction as template DNA. The 20-μL reaction mixture contained 10 μL of HotStarTaq Plus Master Mix 2×, 1 μL of primers (final concentration 0.2 μM), 2 μL of CoralLoad Concentrate 10× (Qiagen, Hilden, Germany), 4 μL of DNase-free water, and 2 μL of DNA. Cycling conditions were as follows: initial denaturation at 95 °C for 5 min, followed by 30 cycles of denaturation at 94 °C for 30 s, annealing at 59 °C for 90 s, extension at 70 °C for 90 s and final extension at 72 °C for 8 min. Ten microliters of PCR product was electrophoresed on 2% agarose gel stained with ethidium bromide (Sigma Aldrich, St. Louis, MO, USA) and PCR products were visualized under ultraviolet light. The sizes of the PCR products were determined by comparison with a molecular size standard (GeneRuler 100 bp Plus DNA ladder; Thermo Fisher Scientific, Waltham, MA, USA). Reference strains of all serotypes obtained from Professor M. Gottschalk (University of Montreal, Montreal, Quebec, Canada) [5,6,7,8] were used as controls in PCR reactions. 

Serotypes identified as 2 or 1/2 were further distinguished by PCR-RFLP method detecting polymorphism in the *cpsK* gene [19]. PCR for the *cpsK* gene was performed in a total volume of 5 µL containing 5 pmol of each primer and PPP Master mix (TopBio, Vestec, Czech Republic). Five microliters of PCR product were digested with 10 U of BstNI restriction endonuclease for 30 min at 60 °C. Resulting DNA fragments were resolved in 2% agarose gel, according to a previous study [19].

Strains not typeable by PCR were serotyped by co-agglutination test. Antisera against all the reference strains were raised in rabbits and co-agglutination reagents were prepared according to previously described coagglutination test [43]. No positive reactions were obtained.

Multilocus sequence typing (MLST) was performed according to previously published method [44]. PCR products of *aroA*, *cpn*, *dpr*, *recA*, *thrA*, *gki* and *mutS* were sequenced by Sanger sequencing method (Eurofins Genomics, Cologne, Germany). PCR for each gene was performed separately in a total volume of 50 µL consisting of 25 pmol of each primer and 1 U of FastStart polymerase (Roche Diagnostics GmbH, Mannheim, Germany) in a recommended buffer. PCR conditions were as follows: initial denaturation at 95 °C for 10 min, followed by 40 cycles of denaturation at 95 °C for 1 min, annealing at 50 °C (for *dpr*, *mutS* and *recA*) or 52 °C (for *cpn* and *thrA*) or 55 °C (for *aroA* and *gki*) for 1 min, elongation at 72 °C for 1 min and final elongation at 72 °C for 7 min. PCR products were verified by gel electrophoresis and subsequently purified by column kit Expin Combo GP (GeneAll Biotechnology Co., LTD, Seoul, South Korea). Allelic identification and sequence type (ST) assignment were done using PubMLST database (https://pubmlst.org/organisms/streptococcus-suis (accessed on 5 October 2021)).

### 4.6. Evolutionary Relationships of Isolates

The evolutionary history was inferred using the neighbor-joining method [45] with the bootstrap test (1000 replicates) [46]. The tree was drawn to scale, with branch lengths in the same units as those of the evolutionary distances used to infer the phylogenetic tree. The evolutionary distances were computed using the maximum composite likelihood method [47] and are in the units of the number of base substitutions per site. This analysis involved 39 nucleotide sequences. All ambiguous positions were removed for each sequence pair (pairwise deletion option). There were a total of 2483 positions in the final dataset. Evolutionary analyses were conducted in MEGA X [48].

### 4.7. Antimicrobial Susceptibility Testing

Antimicrobial susceptibility testing (AST) of ten selected antimicrobials and two combinations of antimicrobials (Table 4) was performed by determination of the minimum inhibitory concentration (MIC) by microdilution and subsequent classification of isolates into susceptibility categories, susceptible, intermediate and resistant, based on clinical breakpoints according to an internationally recognized methodology of the Clinical and Laboratory Standards Institute [49]. The MICs were determined using diagnostic kits manufactured by the co-authors at the Veterinary Research Institute in Brno, Czech Republic. The antimicrobials (Discovery Fine Chemicals Limited, Wimborne, United Kingdom) and their concentrations are listed in Table 4. The growth medium for dilution of antimicrobials is Mueller Hinton Broth (BD Difco, Franklin Lakes, United Kingdom) with the addition of 4% Lysed Horse Blood (Labmediaservis, Jaroměř, Czech Republic). The quality control of the examination was evaluated by parallel examination of the control reference strain *Streptococcus pneumoniae* ATCC 49619 [49].

The 10 antimicrobials and two combinations of antimicrobials used for AST represented 10 antimicrobial groups: fluoroquinolones, phenicols, lincosamides, pleuromutillins, macrolides, tetracyclines, sulfonamides; the penicillins (narrow spectrum, penicillinase sensitive), penicillins with beta-lactamase inhibitors (amoxicillin with clavulanic acid), and cephalosporins (third generation) were considered as three separate groups.

## 5. Conclusions

Diversity of collected isolates with regular occurrence of more than three serotypes per farm was revealed on repopulated farms. The presence of unknown STs and isolates not typable for known serotypes suggests a wide variability of *S. suis* population on Czech large conventional repopulated pig farms.

Our analyses mostly confirmed the unsuitability of lincosamides, tetracyclines and macrolides (often associated with multidrug resistance) for any treatment of *S. suis* infections and, conversely, the suitability of all tested subgroups of penicillin antibiotics, because we did not detect any isolate resistant to them.

Although we were not able to demonstrate an association between short-term antibiotic control treatments and antibiotic resistance found on farms clearly, we share the concerns related to routine antimicrobial treatments, especially related to the perinatal antimicrobial treatment.

## Figures and Tables

**Figure 1 pathogens-10-01314-f001:**
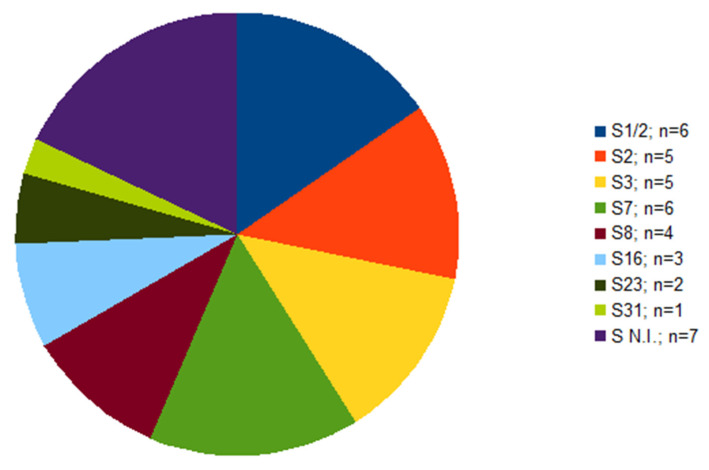
The frequency of serotypes within *S. suis* isolates (n = 39) from diseased pigs. S N.I. = serotype not identified.

**Figure 2 pathogens-10-01314-f002:**
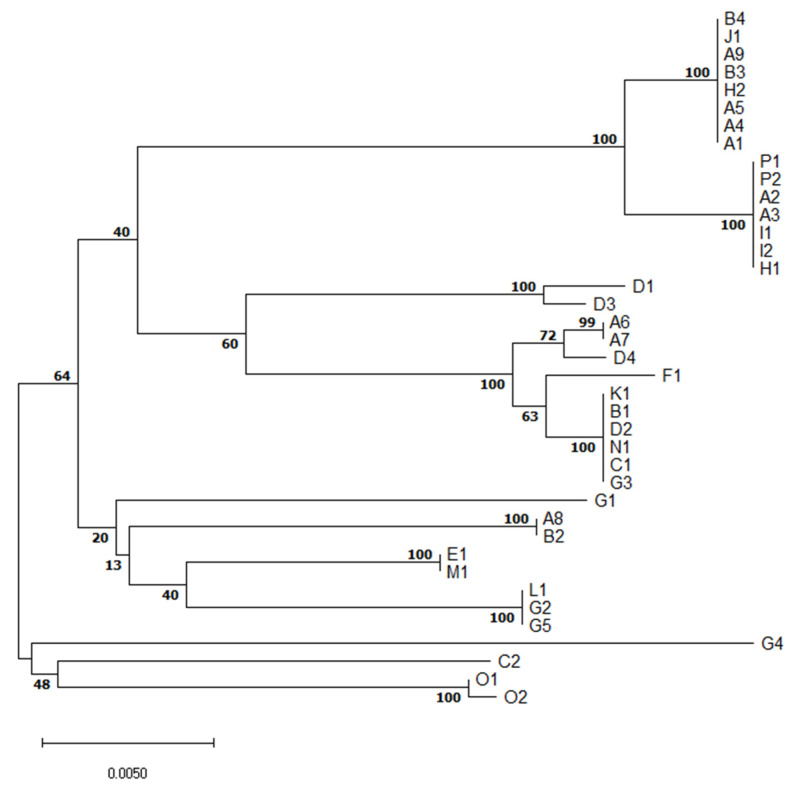
Evolutionary relationships of *S. suis* isolates based on multilocus sequence typing concatenated sequences.The optimal tree with the sum of branch length = 0.15203997 is shown. The percentage of replicate trees in which the associated taxa clustered together in the bootstrap test (1000 replicates) are shown next to the branches. In the A-P farm identification, different numbers denote different isolates of *S. suis* from the same farm.

**Figure 3 pathogens-10-01314-f003:**
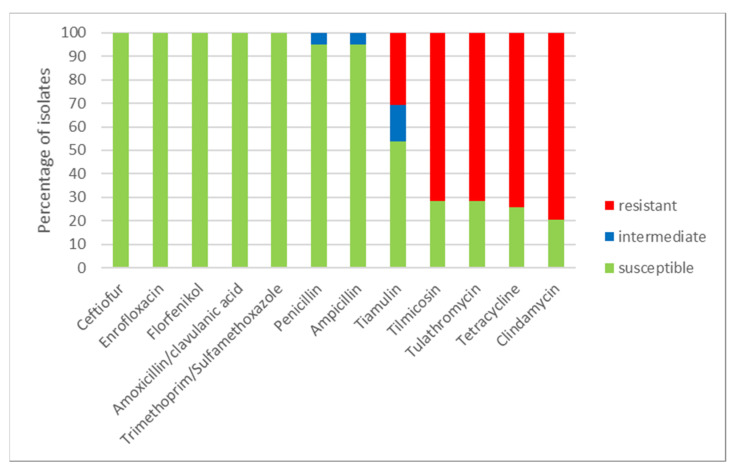
The percentage of susceptible, intermediate and resistant *S. suis* isolates to individual antimicrobials (n = 39).

**Table 1 pathogens-10-01314-t001:** *S. suis* isolates from diseased pigs (n = 39), their serotype, multilocus sequence typing (MLST), organ of isolation and minimum inhibitory concentration (MIC) distribution.

Farm^1^	Date of Sampling	Animal Category	Organ of Isolation	Serotype	MLST	MIC (mg/L)											
						PNC	AMP	AMC *	EFT	ENR	FFC	CLI	TIA	TIL	TUL	TET	SXT **
A1	30.01.2020	weaner	lung	7	29	≤0.03	≤0.03	≤0.25	≤0.125	0.125	2	˃16	32	˃128	˃128	2	≤0.06
A2	15.04.2020	weaner	brain	1/2	28	≤0.03	≤0.03	≤0.25	≤0.125	0.125	1	˃16	0.5	˃128	˃128	˃32	≤0.06
A3	15.04.2020	weaner	pleura	1/2	28	≤0.03	≤0.03	≤0.25	≤0.125	0.25	1	˃16	0.5	˃128	˃128	˃32	≤0.06
A4	05.06.2020	weaner	lung	7	29	≤0.03	≤0.03	≤0.25	≤0.125	0.125	2	˃16	32	˃128	˃128	2	≤0.06
A5	05.06.2020	weaner	lung	7	29	≤0.03	≤0.03	≤0.25	≤0.125	0.125	2	˃16	1	˃128	˃128	2	≤0.06
A6	13.10.2020	weaner	brain	2	54-like, aroA 1	≤0.03	≤0.03	≤0.25	≤0.125	0.125	1	˃16	≤0.25	˃128	˃128	0.5	≤0.06
A7	13.10.2020	weaner	brain	2	54-like, aroA 1	≤0.03	≤0.03	≤0.25	≤0.125	0.25	1	˃16	≤0.25	˃128	˃128	8	≤0.06
A8	13.10.2020	weaner	brain	N.D.	N.D.	≤0.03	≤0.03	≤0.25	≤0.125	0.125	1	˃16	16	˃128	˃128	32	≤0.06
A9	13.11.2020	weaner	brain	7	29	≤0.03	≤0.03	≤0.25	≤0.125	0.125	2	˃16	32	˃128	˃128	2	≤0.06
B1	19.06.2020	weaner	heart	3	54	0.06	0.06	≤0.25	≤0.125	0.25	2	˃16	˃32	˃128	˃128	≤0.25	≤0.06
B2	26.10.2020	weaner	lung	23	N.D.	≤0.03	≤0.03	≤0.25	≤0.125	0.125	1	˃16	16	˃128	˃128	32	≤0.06
B3	26.10.2020	weaner	lung	N.D.	29	≤0.03	≤0.03	≤0.25	≤0.125	0.25	1	≤0.125	1	4	≤1	2	≤0.06
B4	21.01.2021	piglet	joint	7	29	≤0.03	≤0.03	≤0.25	≤0.125	0.25	1	≤0.125	1	4	≤1	2	≤0.06
C1	16.07.2020	weaner	lung	N.D.	54	≤0.03	≤0.03	≤0.25	≤0.125	0.25	1	˃16	˃32	˃128	˃128	≤0.25	≤0.06
C2	26.01.2021	fattener	brain	31	N.D.	0.06	0.06	≤0.25	≤0.125	0.25	2	˃16	˃32	˃128	˃128	32	≤0.06
D1	17.01.2020	weaner	brain	8	1546	≤0.03	≤0.03	≤0.25	≤0.125	0.125	1	˃16	4	˃128	128	32	≤0.06
D2	26.06.2020	weaner	heart	3	54	≤0.03	0.06	≤0.25	≤0.125	0.25	1	˃16	˃32	˃128	˃128	≤0.25	≤0.06
D3	21.08.2020	weaner	lung	3	1521-like, aroA 8	≤0.03	≤0.03	≤0.25	≤0.125	0.125	1	˃16	2	˃128	64	16	≤0.06
D4	27.10.2020	weaner	brain	23	54-like (dpr 223 2 diff. )	≤0.03	≤0.03	≤0.25	≤0.125	0.125	1	˃16	16	˃128	˃128	32	≤0.06
E1	27.01.2020	weaner	lung	2	1	≤0.03	≤0.03	≤0.25	≤0.125	0.25	1	4	˃32	4	≤1	32	≤0.06
F1	28.01.2020	weaner	lung	N.D.	N.D.	≤0.03	≤0.03	≤0.25	≤0.125	0.125	1	˃16	≤0.25	˃128	˃128	≤0.25	≤0.06
G1	22.01.2020	weaner	eye	16	1222-like (gki 294 1 diff.)	0.25	0.06	≤0.25	0.5	0.25	1	8	16	2	≤1	32	≤0.06
G2	20.08.2020	weaner	lung	8	87	≤0.03	≤0.03	≤0.25	≤0.125	0.25	1	≤0.125	0.5	2	≤1	0.5	1
G3	01.09.2020	weaner	lung	3	54	≤0.03	0.06	≤0.25	≤0.125	0.5	1	8	˃32	˃128	˃128	≤0.25	≤0.06
G4	01.09.2020	weaner	lung	N.D.	912	≤0.03	≤0.03	≤0.25	≤0.125	0.125	1	1	2	2	≤1	32	≤0.06
G5	01.09.2020	weaner	lung	8	87	≤0.03	≤0.03	≤0.25	≤0.125	0.125	1	≤0.125	0.5	2	≤1	0.5	1
H1	07.10.2020	weaner	brain	2	28	≤0.03	0.06	≤0.25	≤0.125	0.125	1	≤0.125	1	4	≤1	32	≤0.06
H2	07.10.2020	weaner	lung	N.D.	29	≤0.03	≤0.03	≤0.25	≤0.125	0.125	1	≤0.125	1	4	≤1	32	≤0.06
I1	18.08.2020	weaner	lung	1/2	28	≤0.03	≤0.03	≤0.25	≤0.125	0.125	1	˃16	1	˃128	˃128	32	≤0.06
I2	18.08.2020	weaner	heart	1/2	28	≤0.03	≤0.03	≤0.25	≤0.125	0.125	1	˃16	0.5	˃128	˃128	32	≤0.06
J1	09.02.2021	piglet	brain	7	29	≤0.03	≤0.03	≤0.25	≤0.125	0.125	2	˃16	32	˃128	˃128	2	≤0.06
K1	17.02.2020	piglet	lung	3	54	≤0.03	0.06	≤0.25	≤0.125	0.25	1	˃16	˃32	˃128	˃128	≤0.25	≤0.06
L1	10.02.2020	fattener	lung	8	87	≤0.03	≤0.03	≤0.25	≤0.125	0.25	1	˃16	0.5	˃128	˃128	˃32	≤0.06
M1	10.11.2020	fattener	pericardium	2	1	≤0.03	≤0.03	≤0.25	≤0.125	0.25	2	˃16	0.5	˃128	˃128	32	≤0.06
N1	26.06.2020	fattener	lung	N.D.	54	≤0.03	0.06	≤0.25	≤0.125	0.25	1	˃16	˃32	˃128	˃128	≤0.25	≤0.06
O1	12.11.2020	piglet	lung	16	1280-like (mutS 139 1 diff.)	0.5	1	≤0.25	1	0.25	1	˃16	16	˃128	˃128	˃32	0.5
O2	12.11.2020	piglet	lung	16	1280-like (mutS 139 1 diff.)	0.5	1	≤0.25	1	0.25	1	˃16	16	˃128	˃128	˃32	0.5
P1	23.01.2020	piglet	joint	1/2	28	≤0.03	≤0.03	≤0.25	≤0.125	0.125	1	≤0.125	1	4	2	˃32	≤0.06
P2	23.01.2020	piglet	joint	1/2	28	≤0.03	≤0.03	≤0.25	≤0.125	0.25	1	≤0.125	1	4	2	˃32	≤0.06

^1^ A-P farm identification. Different numbers denote different isolates of *S. suis* from the same farm. N is a finishing farm of piglets with origin on farm E. P is the farm in Slovakia. * Concentration in the table is for amoxicillin. ** Concentration in the table is for trimethoprim. PEN = penicillin; AMP = ampicillin; AMC = amoxicillin/clavulanic acid 2/1; EFT = ceftiofur; ENR = enrofloxacin; FFC = florfenicol; CLI = clindamycin; TIA = tiamulin; TIL = tilmicosin; TUL = tulathromycin; TET = tetracycline; SXT = trimethoprim/sulfamethoxazole 1/19; N.D. = not determined; green cells = susceptible; blue cells = intermediate; red cells = resistant.

**Table 2 pathogens-10-01314-t002:** Resistance ^1^ profiles for *S. suis* isolates (n = 39).

Frequency of Resistance by		Phenotype of Resistance	Number of Isolates	Number of Multidrug-Resistant Isolates
Active Substance	Antimicrobial Group			
0	0		2	
1	1	TET	6	
2	2	CLI, TET	2	
3	2	CLI, TIL, TUL	2	0
3	3	CLI, TIA, TET	1	
4	3	CLI, TIL, TUL, TET	15	
4	3	CLI, TIA, TIL, TUL	6	
5	4	CLI, TIA, TIL, TUL, TET	5	27

^1^ Intermediate isolates were not considered resistant. TET = tetracycline; CLI = clindamycin; TIL = tilmicosin; TUL = tulathromycin; TIA = tiamulin.

**Table 3 pathogens-10-01314-t003:** Pig farm and *S. suis* isolate characteristics and used routine antibiotic control treatments.

Farm	Sows (Framework Number)	Antimicrobials Used in		*S. suis* Isolates (n)	Isolates Resistant to TUL (n)	Multidrug Resistant Isolates (n)
		Piglets	Weaners			
A	2000	TUL	AMO	9	9	8
B	2000	TUL	-	4	2	2
C	2000	TUL	-	2	2	2
D	2000	TUL	AMO	4	4	4
E	700	TUL	AMO	1	0	1
F	600	TUL	AMO	1	1	0
G	1000	TUL	AMO	5	1	1
H	1500	-	AMO	2	0	0
I	1100	TUL	AMO	2	2	2
J	1700	-	AMO	1	1	1
K	600	TUL	AMO	1	1	1
L	1500	-	AMO	1	1	1
M	500	-	AMO	1	1	1
N	-	TUL	AMO	1	1	1
O	500	-	AMO	2	2	2
P	2000	-	AMO	2	0	0
Total	19,700	10/16	14/16	39 (30 ^1^; 9 ^2^)	28 (23 ^1^; 5 ^2^)	27 (22 ^1^; 5 ^2^)

TUL = tulathromycin; AMO = amoxicillin; P = farm located in Slovakia; ^1^ farms using routine treatment with tulathromycin in the first week of life; ^2^ farms not using routine treatment with tulathromycin.

**Table 4 pathogens-10-01314-t004:** Range of concentrations of antimicrobials in the kit for antimicrobial susceptibility testing of *S. suis* isolates with the clinical breakpoints.

PEN	AMP	AMC *	EFT	ENR	FFC	CLI	TIA	TIL	TUL	TET	SXT **
4	4	32	16	8	64	16	32	128	128	32	PC
2	2	16	8	4	32	8	16	64	64	16	4
1	1	8	4	2	16	4	8	32	32	8	2
0.5	0.5	4	2	1	8	2	4	16	16	4	1
0.25	0.25	2	1	0.5	4	1	2	8	8	2	0.5
0.125	0.125	1	0.5	0.25	2	0.5	1	4	4	1	0.25
0.06	0.06	0.5	0.25	0.125	1	0.25	0.5	2	2	0.5	0.125
0.03	0.03	0.25	0.125	0.06	0.5	0.125	0.25	1	1	0.25	0.06

Green cells = susceptible; blue cells = intermediate; red cells = resistant. The concentrations of antimicrobials in the table are given in mg/l. * concentration in the table is for amoxicillin; ** concentration in the table is for trimethoprim; PEN = penicillin; AMP = ampicillin; AMC = amoxicillin/clavulanic acid 2/1; EFT = ceftiofur; ENR = enrofloxacin; FFC = florfenicol; CLI = clindamycin; TIA = tiamulin; TIL = tilmicosin; TUL = tulathromycin; TET = tetracycline; SXT = trimethoprim/sulfamethoxazole 1/19; PC = positive growth control.

## Data Availability

The data presented in this study are available on request from the corresponding author. The more detailed data are not publicly available to protect the privacy of farm owners.

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
