# Peer review of "Streptococcus suis Isolates—Serotypes and Susceptibility to Antimicrobials in Terms of Their Use on Selected Repopulated Czech Pig Farms"

_pathogens, 2021, doi:10.3390/pathogens10101314_

Round 1

Reviewer 1 Report

Well-structured and reported study on the variability of S, suis infection and strains dynamics on field level. A few points need to be addressed. Comments are demonstrated in the pdf file.

Author Response

Dear Editor,

We would like to thank you and both reviewers for helpful and encouraging comments. According to recommendation of reviewer 2 we performed phylogenetic analysis based on MLST sequences and add this analysis to the manuscript. Changed parts were tracked. Pleas find detailed response to all recommendations and comments below.

Ján Matiašovic

Reviewer 1

Well-structured and reported study on the variability of S, suis infection and strains dynamics on field level. A few points need to be addressed.

line 47

to which commercial vaccines the authors refer? See also line 74

Response: Thank you for this note. We would like to stress, that preparation of effective vaccine against S. suis is difficult and in fact no commercial vaccine (i.e. vaccine authorized as veterinary medicinal product with fixed composition of the serotypes involved) is available. In principle there is possible to produce autogenous vaccine tailored for the epidemiological situation in epidemiologically related units, in line with rules as stipulated by national legal provisions of individual Member States. The word “commercial” was removed from both sentences.

line 99

Was such decision based on specific pathogen effects? At which extent is repopulation financially better than treatment of disease?

Response: We apologize for the vague wording. We would like to say, that repopulation is the technique used in some cases to eliminate particular pathogen. It may be efficient and effective in the case of some bacterial pathogens as is A. pleuropneumoniae etc.. We believe the rephrased sentence is more clear.

line 105-106

move to results

Response: Thank you for this comment, the three sentences were removed.

line 119

Table should be self explanatory. Pleare add "multilocus sequence typing " before MLST abbreviation in the title.

Response: Thank you for this comment, we add "multilocus sequence typing " before MLST abbreviation in the title.

line 137-143

Please avoid paragraphs with one sentence.

Response: Sentences were joined to one paragraph.

line 197 - Discussion

Are there data to compare S.suis occurence and resistance patterns prior and after repopulation of the tested farms? Did repopulation strategy had any effect on S. suis strains and AMR?

Response: This is an interesting question. Unfortunately, although many farms included in the study were observed for many years before repopulation, S. suis was not identified in clinical samples from sick pigs. The farms had problems with other pathogens. Therefore, nothing can be said about the occurrence and resistance of S. suis before repopulation and the effect of repopulations on S. susis strains and AMR. According to reviewer 2 reccomendation, the paragraph was removed.

line 299-301

Please avoid paragraphs with one sentence.

Response: Sentences were joined to one paragraph.

line 310

Were both testings performed in the same year (2020)?

Response: Thank you for this comment, the sentence was rewritten, also by adding comparison of tulathromycin-resistance recorded in NAP for the years 2017-2020. The reliability of the information that the number of TUL-resistant S. suis strains on the studied farms is higher than on other farms in the Czech Republic with different management has increased.

line 311-313

Please avoid paragraphs with one sentence.

Response: Sentences were joined to one paragraph.

line 315

Response: Thank you for this suggestion, the word “prudent” was added.

line 351

Response: Thank you for this suggestion, the word “reared” was added.

line 375

way of euthanization?

Response: Thank you for this comment, the information was added to the text. The authorized veterinary medicinal product T61 was administered to anesthetized animals, according to T61 approved summary of product characteristics (SPC).

line 378

Response: We are sorry to forget to mention manufacturer. The information was added to the text.

line 383

name , place of the lab?

Response: “The samples were subjected to conventional bacteriology (cultivation, isolation and identification of the agents) in an accredited diagnostic laboratory Sevaron, s.r.o., Brno, Czech Republic.”

line 405

according to a previous study

Response: Thank you for this suggestion, the “according to a previous study” was used in the sentence.

line 427

Please add this sentence to the end of the precious paragraph.

Response: Thank you for this suggestion, the sentence was joined with previous paragraph.

line 435

acoording to a previous study

Response: Thank you for this suggestion, the “according to a previous study” was used in the sentence.

Reviewer 2 Report

The presented study provides epidemiological data on prevalence of Streptococcus suis serotypes and sequence types as well as their antibiotic resistance profile in the Czech Republic. The study design is sound and the results are mostly presented well. It is certainly of relevance to the field that formerly unknown ST were detected and their spread and virulence should be monitored in the future. It is also noteworthy that perinatal treatment seems likely linked to tulathromycin resistance in older pigs on the same farm. This observation should be continued to be studies by the authors.

My main concern is the small number of isolates used in the study. Even though isolates originated from 16 different farms, 39 isolates in total is too small of a number to draw any real conclusions. I suggest the authors consider increasing their sample number and waiting with submission of a manuscript until they have reached a total number of samples that allows statistical analysis of their data which would greatly strengthen the merit of the manuscript. At this point, the data is still worthy of publication but does not give a representative image of the S. suis situation in the CZ. Maybe there are more isolates available from previous years that could be included in the study since the authors state that systematic antibiotic resistance monitoring including S. suis started in CZ in 2016?

Additionally, here are some improvements that need to be made to the manuscript:

The authors state that they isolated several strains that could not be assigned to a certain serotype by PCR. Did the authors consider subjecting these strains to classical serotyping by latex agglutination?

lines 97-100: it is not clear what that sentence means. Please rephrase. Also, why did this repopulation take place and what did it entail? The authors explain it better in the discussion (lines 228 following) but it would be useful to understand "repopulation" earlier on.

Can you really say that S 1/2 and 7 are the most frequently isolated ones (n=6) if they differ from less frequently isolated ones (n=1-4) by only 5 to 2 isolates. Consider rephrasing your conclusions or including more samples. 

Presenting the MLST data as a dendrogram would be very helpful.

Consider presenting data in Fig. 2 as % of isolates rather than absolute numbers.

Paragraph 2.4 does not provide any relevant information and it seems like the authors were forcing to find correlations. Either eliminate the paragraph or finish it with a conclusion of what all these results mean.

line 228: was "exceptionally used" is probably not the English term the authors were looking for. Did you mean rarely?

line 266-267: The reader is misled to believe that the sentence refers to the data from the NAP. It should be clarified that the statement refers to data from Spain and Brazil.

line 274: needs a citation

line 425: please list the reference strains that were used

Author Response

Dear Editor,

We would like to thank you and both reviewers for helpful and encouraging comments. According to recommendation of reviewer 2 we performed phylogenetic analysis based on MLST sequences and add this analysis to the manuscript. Changed parts were tracked. Pleas find detailed response to all recommendations and comments below.

Ján Matiašovic

Reviewer 2

The presented study provides epidemiological data on prevalence of Streptococcus suis serotypes and sequence types as well as their antibiotic resistance profile in the Czech Republic. The study design is sound and the results are mostly presented well. It is certainly of relevance to the field that formerly unknown ST were detected and their spread and virulence should be monitored in the future. It is also noteworthy that perinatal treatment seems likely linked to tulathromycin resistance in older pigs on the same farm. This observation should be continued to be studies by the authors.

My main concern is the small number of isolates used in the study. Even though isolates originated from 16 different farms, 39 isolates in total is too small of a number to draw any real conclusions. I suggest the authors consider increasing their sample number and waiting with submission of a manuscript until they have reached a total number of samples that allows statistical analysis of their data which would greatly strengthen the merit of the manuscript. At this point, the data is still worthy of publication but does not give a representative image of the S. suis situation in the CZ. Maybe there are more isolates available from previous years that could be included in the study since the authors state that systematic antibiotic resistance monitoring including S. suis started in CZ in 2016?

Response: Thank you for this comment, we add comparison of tulathromycin-resistance recorded in NAP for the years 2017-2020. The reliability of the information that the number of TUL-resistant S. suis strains on the studied farms is higher than on other farms in the Czech Republic with different management has increased. Unfortunately, with the linear continuation of the development of the situation on the monitored farms and with the linear continuation of the current data collection, we would need another two years of data collection for the differences to reach statistical significance purely within our own data. However, for the reason that even farms that do not use TUL perinatally can probably be burdened by its use on farms supplying them with breeding material, the comparison of our entire set with the NAP results is in a way "cleaner" and easier to interpret. However, we also consider important to publish our at least such substantiated doubts about the suitability of current routine practice now (and not in a few years), because it is highly topical to support studied farms away from reliance on perinatal routine ATB administration and direct them for other measures. Although this task is very complex and challenging issue.

Additionally, here are some improvements that need to be made to the manuscript:

The authors state that they isolated several strains that could not be assigned to a certain serotype by PCR. Did the authors consider subjecting these strains to classical serotyping by latex agglutination?

Response: Thank you for this comment. We add the following information:

Strains non typeable by PCR were serotyped by co-agglutination test. Antisera against all the reference strains were raised in rabbits and co-agglutination reagents were prepared according to Mittal et al, 1983. No positive reaction were obtained.”

lines 97-100: it is not clear what that sentence means. Please rephrase. Also, why did this repopulation take place and what did it entail? The authors explain it better in the discussion (lines 228 following) but it would be useful to understand "repopulation" earlier on.

Response: Thank you for this comment. We would like to say, that repopulation is the technique used in some cases to eliminate particular pathogen. It may be efficient and effective in the case of some bacterial pathogens as is A. pleuropneumoniae etc.. The SPF system can be cost-effective: less medication is needed and vaccination costs are reduced. It does depend on strict biosecurity and a closed herd policy or strict sourcing and transportation from controlled herds with similar health status. The impact of the SPF system will be greatest in farms towards the top of the breeding pyramid (EMA and EFSA (2017), European Medicines Agency and European Food Safety Authority: Joint Scientific Opinion on measures to reduce the need to use antimicrobial agents in animal husbandry in the European Union, and the resulting impacts on food safety (RONAFA). EFSA Journal 15(1):4666). We believe the rephrased sentence is clearer.

Can you really say that S 1/2 and 7 are the most frequently isolated ones (n=6) if they differ from less frequently isolated ones (n=1-4) by only 5 to 2 isolates. Consider rephrasing your conclusions or including more samples. 

Response: Thank you for this note, the sentence was rephrased.

Presenting the MLST data as a dendrogram would be very helpful.

Response: Thank you for this suggestion, the phylogenetic tree was constructed from concatenated MLST sequences.

Consider presenting data in Fig. 2 as % of isolates rather than absolute numbers.

Response: Thank you for this suggestion, the Fig. 2 was modified to show percentages.

Paragraph 2.4 does not provide any relevant information and it seems like the authors were forcing to find correlations. Either eliminate the paragraph or finish it with a conclusion of what all these results mean.

Response: Thank you for this suggestion, the Paragraph 2.4 was deleted.

line 228: was "exceptionally used" is probably not the English term the authors were looking for. Did you mean rarely?

Response: Thank you for this suggestion, the word “rarely” was used.

line 266-267: The reader is misled to believe that the sentence refers to the data from the NAP. It should be clarified that the statement refers to data from Spain and Brazil.

Response: Thank you for this suggestion, the sentence was rephrased: “According to previously published data from Spain and Brazil, only low percentages...…”

line 274: needs a citation

Response: Thank you for this note, the EMA 2019 citation [42] was added.

line 425: please list the reference strains that were used

Response: Thank you for this suggestion, we rephrased sentence and added citations for reference strains used.